# A Novel Predictive Model for In-Hospital Mortality Based on a Combination of Multiple Blood Variables in Patients with ST-Segment-Elevation Myocardial Infarction

**DOI:** 10.3390/jcm9030852

**Published:** 2020-03-20

**Authors:** Yuhei Goriki, Atsushi Tanaka, Kensaku Nishihira, Atsushi Kawaguchi, Masahiro Natsuaki, Nozomi Watanabe, Keiichi Ashikaga, Nehiro Kuriyama, Yoshisato Shibata, Koichi Node

**Affiliations:** 1Miyazaki Medical Association Hospital Cardiovascular Center, Miyazaki 880-0834, Japan; i03.eou2@gmail.com (Y.G.); nishihira@med.miyazaki-u.ac.jp (K.N.); n_watanabe@cure.or.jp (N.W.); ashikaga@cure.or.jp (K.A.); n-kuriyama@cure.or.jp (N.K.); yshibata@cure.or.jp (Y.S.); 2Department of Cardiovascular Medicine, Saga University, Saga 849-8501, Japan; natsuaki@kuhp.kyoto-u.ac.jp (M.N.); node@cc.saga-u.ac.jp (K.N.); 3Clinical Research Center, Saga University Hospital, Saga 849-8501, Japan; akawa@cc.saga-u.ac.jp

**Keywords:** ST-segment-elevation myocardial infarction, risk score, in-hospital mortality

## Abstract

In emergency clinical settings, it may be beneficial to use rapidly measured objective variables for the risk assessment for patient outcome. This study sought to develop an easy-to-measure and objective risk-score prediction model for in-hospital mortality in patients with ST-segment elevation myocardial infarction (STEMI). A total of 1027 consecutive STEMI patients were recruited and divided into derivation (*n* = 669) and validation (*n* = 358) cohorts. A risk-score model was created based on the combination of blood test parameters obtained immediately after admission. In the derivation cohort, multivariate analysis showed that the following 5 variables were significantly associated with in-hospital death: estimated glomerular filtration rate <45 mL/min/1.73 m^2^, platelet count <15 × 10^4^/μL, albumin ≤3.5 g/dL, high-sensitivity troponin I >1.6 ng/mL, and blood sugar ≥200 mg/dL. The risk score was weighted for those variables according to their odds ratios. An incremental change in the scores was significantly associated with elevated in-hospital mortality (*p* < 0.001). Receiver operating characteristic curve analysis showed adequate discrimination between patients with and without in-hospital death (derivation cohort: area under the curve (AUC) 0.853; validation cohort: AUC 0.879), and there was no significant difference in the AUC values between the laboratory-based and Global Registry of Acute Coronary Events (GRACE) score (*p* = 0.721). Thus, our laboratory-based model might be helpful in objectively and accurately predicting in-hospital mortality in STEMI patients.

## 1. Introduction

Acute myocardial infarction (AMI) remains a worldwide leading cause of high mortality [1]. Over the last 2 decades, advances in the coronary care unit and primary reperfusion therapy have improved the outcome of the chronic phase after an AMI [2]. Although the incidence of in-hospital death in patients with AMI also declined up to the first decade of the twenty-first century [3,4], thereafter improvement in mortality appeared to plateau [5]. Intriguingly, the incidence of in-hospital death in Japan has been significantly higher in patients with ST-segment elevation myocardial infarction (STEMI) than in patients with non-STEMI (7.7% vs. 5.1%), despite the recently introduced settings of optimal medical therapy and successful reperfusion after primary percutaneous coronary intervention (PCI) [6]. Additionally, the shortening of door-to-balloon times in patients with STEMI was not associated with improved short- and long-term clinical outcomes, which suggests a need for additional strategies [7,8]. Thus, the increased accuracy of prompt risk stratification in the emergency department might aid in improving the outcome of patients with AMI during the acute phase, especially in patients with STEMI. Accurate risk stratification should raise an important implication in the management of these patients.

The Thrombolysis in Myocardial Infarction (TIMI) risk score and the Global Registry of Acute Coronary Events (GRACE) risk score have been reported to be useful risk stratification tools for in-hospital mortality in patients with AMI [9,10]. Of these risk scores, the GRACE risk score is known to have the highest validity [11,12]. The GRACE risk score requires 8 variables to assess risk of in-hospital mortality, as follows: age, heart rate, systolic blood pressure, Killip classification, cardiac arrest at hospital admission, initial serum creatinine level, elevated levels of cardiac markers, and ST-segment deviation. However, the heart rate and systolic blood pressure sometimes vary widely during the acute phase, and the Killip classification is a subjective evaluation. Therefore, performing an assessment of these variables in the emergency setting sometimes proves difficult. By contrast, blood parameters can be quickly measured and provide objective information, even in the emergency setting. Several biomarkers have been reported to be potential tools for predicting in-hospital mortality in patients with AMI [13,14,15,16,17,18,19,20]. Given that these biomarkers can reflect different aspects of pathophysiological responses that occur during the post-AMI phase, a combination of biomarkers might provide more accurate and useful information for risk stratification than the information provided by any individual biomarker. Therefore, the aim of this study was to develop a novel, easy-to-measure, and objective risk-score prediction model for in-hospital mortality in patients with STEMI that was based on a combination of parameters obtained on routine blood tests, and to compare the predictive utility of the new model with that of the conventional GRACE score.

## 2. Methods

### 2.1. Study Design and Population

This was a single-center, retrospective observational study undertaken in Japan. A total of 1252 consecutive patients who were admitted to Miyazaki Medical Association Hospital for STEMI between April 2012 and May 2019 were enrolled in the study. Exclusion criteria included the following: (1) not receiving primary PCI; (2) onset-to-admission time >48 h; (3) lack of blood test results. A total of 1027 patients were finally included in the study. Based on the date of hospital admission, they were divided into 2 groups, the derivation and validation sets, which consisted of 669 patients hospitalized from April 2012 to December 2016 and 358 patients hospitalized from January 2017 to May 2019, respectively (Figure 1). The protocol of the study was approved by the Institutional Review Board at Miyazaki Medical Association Hospital (2019–30).

### 2.2. Diagnosis of STEMI

STEMI was diagnosed according to the universal definition of MI described by the European Society of Cardiology and the American Heart Association [21]. After urgent admission to our hospital and within 48 h after onset of symptoms, all the patients who were suspected to have STEMI, according to the clinical manifestations such as changes in the electrocardiogram and elevated cardiac enzymes, received emergency coronary angiography and subsequent coronary revascularization. Reperfusion therapy was performed along with primary PCI according to the relevant guidelines and recommendation [1]. The patients then received optimal medications.

### 2.3. Data Collection

The following types of data were collected: demographic characteristics of study patients, medical history, presenting signs and symptoms, results of blood tests, transthoracic echocardiographic and electrocardiographic findings, cardiac procedures, and in-hospital outcome. 

Transthoracic echocardiography was performed for all patients immediately after admission, and left ventricular ejection fraction (LVEF) was estimated by the standard biplane Simpson method. The white blood cell (WBC) count (Sysmex XN-1000^TM^ (Sysmex Corporation, Hyogo, Japan)); platelet count (Sysmex XN-1000^TM^ (Sysmex Corporation, Hyogo, Japan)); and levels of hemoglobin (Sysmex XN-1000^TM^ (Sysmex Corporation, Hyogo, Japan)), C-reactive protein (CRP, BioMajesty^TM^ Series JCA-BM6010 (JEOL Ltd., Tokyo, Japan)), creatinine (BioMajesty^TM^ Series JCA-BM6010 (JEOL Ltd., Tokyo, Japan)), creatine kinase (CK, BioMajesty^TM^ Series JCA-BM6010 (JEOL Ltd., Tokyo, Japan)), blood sugar (BS, BioMajesty^TM^ Series JCA-BM6010 (JEOL Ltd., Tokyo, Japan)), HbA1c (HLC-723^®︎^G9 analyzer (Tosho Bioscience, Tokyo, Japan), albumin (BioMajesty^TM^ Series JCA-BM6010 (JEOL Ltd., Tokyo, Japan)), uric acid (BioMajesty^TM^ Series JCA-BM6010 (JEOL Ltd., Tokyo, Japan)), low-density lipoprotein cholesterol (LDL-CHO, BioMajesty^TM^ Series JCA-BM6010 (JEOL Ltd., Tokyo, Japan)), high-density lipoprotein cholesterol (HDL-CHO, BioMajesty^TM^ Series JCA-BM6010 (JEOL Ltd., Tokyo, Japan)), high-sensitivity troponin I (hsTnI, ARCHITECT^®^ high sensitive troponin I (Abbott Japan, Tokyo, Japan) on an ARCHITECT^®^ i1000SR analyzer (Abbott Japan, Tokyo, Japan)), and brain natriuretic peptide (BNP, AIA^®^-900 analyzer (Tosho Bioscience, Tokyo, Japan)) were measured in blood specimens obtained immediately after admission. The estimated glomerular filtration rate (eGFR) was calculated using the revised equation for the Japanese population [22]. 

### 2.4. Statistical Analysis

Continuous variables are reported as means ± standard deviation for normally distributed values and as medians (interquartile range) for non-normal values. Categorical variables are expressed as numbers and percentages. Comparisons of continuous variables between groups were performed by the Student t-test or Mann–Whitney U test, as appropriate. Comparisons of categorical variables were assessed by the chi-squared or Fisher exact test, as appropriate. Univariate logistic regression analysis was used to calculate the effects of multiple variables on in-hospital death. Potential risk markers were eliminated by multivariate logistic regression using stepwise factor elimination. An odds ratio (OR) was obtained for each significant variable. Based on the OR obtained by multivariate logistic regression, the risk factors for in-hospital death were assigned weighted integers (OR < 3 = 1 point, OR ≥ 3 = 2 points). The patients were then classified into 3 groups according to the risk score, as follows: low-risk (0 to 1 point), moderate-risk (2 to 4 points), and high-risk group (≥5 points). Receiver operating characteristic curve analysis was used to evaluate the effectiveness of the risk score for predicting in-house mortality, and the area under the curve (AUC) was used to determine the predictability of the risk score. We calculated the AUC of the GRACE score and compared it with that of the risk score. The analyses were performed by the JMP software program, version 14.2.0 (SAS Institute Inc., Cary, NC, USA). *p*-values < 0.05 were considered statistically significant. 

## 3. Results

### 3.1. Patient Characteristics

A total of 1027 patients (669 in the derivation cohort and 358 in the validation cohort) were enrolled in this analysis. Table 1 shows the patients’ admission characteristics as stratified into study cohorts. No significant differences were seen between the derivation and validation cohorts for the clinical parameters, with the exception of smoking history; or for vital signs or laboratory data. No significant differences in the clinical parameters related to the treatments for STEMI were observed between the 2 cohorts. Overall, 57 (5.6%) in-hospital deaths were observed in the study. There was no significant difference in the onset-to-admission time between in-hospital survival and death groups in the overall cohort (survivor, median 200 min (interquartile range 115–385) vs. death, 260 min (145–630), *p* = 0.062). 

### 3.2. Blood Testing and Risk Stratification Model

Table 2 shows the univariate analysis of the results of blood testing in the derivation cohort stratified by in-hospital survival or death. The variables that were significant by univariate analysis were subjected to multivariate stepwise forward logistic regression analysis, and 5 variables were found to be significantly associated with in-hospital death, as follows: platelet count <15×10^4^ (OR 3.45, 95% confidence interval (CI) 1.50–7.97; *p* = 0.003), BS ≥200 mg/dL (OR 2.63, 95% CI 1.20–5.80; *p* = 0.020), eGFR <45 mL/min/1.73m^2^ (OR 3.65, 95% CI 1.80–8.11; *p* = 0.001), albumin ≤3.5g/dL (OR 3.37, 95% CI 1.52–7.47; *p* = 0.003), and hsTnI >1.6 ng/mL (OR 2.76; 95% CI 1.27–6.01; *p* = 0.010). Based on their OR values, risk scores were assigned to the 5 variables (Table 3).

### 3.3. Prediction of In-Hospital Mortality

No significant differences were found between the in-hospital mortality of the 2 patient cohorts (5.1% for derivation cohort vs. 6.4% in the validation cohort, *p* = 0.318). In the derivation cohort, an increased total risk score was significantly associated with elevated in-hospital mortality (*p* < 0.001) (Figure 2A). Therefore, we assessed the utility of this risk score in the validation cohort. The risk score also showed a significant trend for in-hospital mortality in the validation cohort (*p* < 0.001) (Figure 2B). Additionally, we classified the patients into 3 groups according to risk score to simplify its use in clinical settings. Low-risk group (0 to 1 point), moderate-risk group (2 to 4 points), high-risk group (≥ 5 points). These risk groups also showed a significant trend for in-hospital mortality among the respective validation and derivation cohorts (low-risk 1.4% and 0.8%, moderate-risk 8.8% and 12.2%, and high-risk 29.8% and 35.8%; *p* < 0.001) (Figure 3). The risk score displayed adequate discrimination between patients with or without in-hospital death in both the validation (AUC: 0.853, 95% CI 0.782–0.904) and derivation cohorts (AUC: 0.879, 95% CI 0.791–0.933) (Figure 4).

### 3.4. Comparison of New Risk Score with GRACE Risk Score

The AUCs of the laboratory risk score and GRACE risk score in the validation cohort were 0.879 (95% CI 0.791–0.933) and 0.891 (95% CI 0.773–0.952), respectively (Figure 5A). The difference between the AUCs was not significant (*p* = 0.721). Furthermore, when patients in the 3 laboratory risk-score groups (0–1, 2–4, and ≥5) were subdivided into the low-intermediate (GRACE risk score <155) or high-risk (≥155) subgroups, [23] our laboratory-based risk model could further stratify the risk of in-hospital mortality even in the high-risk patients identified by the GRACE risk score, as well as in the low-intermediate risk patients (Figure 5B).

## 4. Discussion

The major findings of this study of patients with STEMI undergoing primary PCI were as follows: (1) each of 5 laboratory parameters quantified at admission (eGFR <45 mL/min/1.73m^2^, platelet count <15 × 10^4^/μL, albumin level ≤3.5 g/dL, hsTnI level >1.6 ng/mL, and BS level ≥200 mg/dL) was independently associated with increased in-hospital mortality; (2) the risk-weighted combined-score model based on these laboratory parameters could incrementally provide an accurate prediction of in-hospital mortality; (3) the predictive value of this model for in-hospital mortality was comparable to that of the conventional GRACE risk score model, and our model could further stratify that risk, especially for high-risk patients as identified by the GRACE model. Thus, our results suggest that this laboratory-based risk model can enable the objective and accurate prediction of the risk of in-hospital mortality in STEMI patients who underwent primary PCI within 48 h after onset.

All patients with STEMI seen in the emergency department are required to receive a prompt risk assessment of cardiovascular complications and the short-term prognosis [1]. In this study, we focused on the results of routine blood tests to develop an objective risk score model for predicting in-hospital mortality in patients with STEMI. Several established risk models, including the TIMI and GRACE risk scores, use vital signs and the Killip classification in their evaluations [9,10]. However, those variables often vary widely, especially in the acute clinical setting; and Killip classification is a subjective evaluation based on the physical examination. The patient’s retrospective clinical history is also needed for these models, and it can be difficult to obtain from patients with such clinical conditions as shock and cardiac arrest. By contrast, the results of blood tests can be quickly and easily obtained, even in the emergency setting, and will provide completely objective information. Biomarkers can be also time-varying during acute presentation of STEMI. However, there was no significant difference in the onset-to-admission time between in-hospital survival group and death group in our study, suggesting the higher levels of biomarkers in the death group were at least unlikely due to later presentation and/or sampling. Given the study findings that our laboratory-based risk score model could provide adequate discrimination between patients with and without in-hospital death that is comparable to that provided by the GRACE risk score, our model should help us provide a more objective assessment than that provided by the GRACE model for the short-term risk of mortality in STEMI patients after successful primary PCI. 

Several biomarkers that are related to cardiac, metabolic, hematologic, and inflammatory responses have been reported to be independent predictors of in-hospital mortality in patients with STEMI [13,14,15,16,17,18,19,20]. Our laboratory-based risk model was composed of 5 variables (platelet count and levels of BS, albumin, eGFR, and hsTnI) obtained from routine admission blood tests. Each of these variables has also been demonstrated to be independently associated with increased risk of poor prognosis in patients with acute coronary syndrome (ACS), including STEMI [24,25,26,27,28]. These biomarkers are likely associated with not only acute pathophysiological responses such as inflammation and cardiac direct injury that resulted from the development of an AMI, but also with the patient’s general conditions, including metabolic conditions and nutrition. Interestingly, an elevated level of BS, but not HbA1c, was associated with mortality in both univariate and multivariate analyses. This finding suggests that hyperglycemia possibly as a reflection of an acute stress response, rather than underlying and longer-term diabetes status which is a well-known risk factor for development of AMI, is associated with short-term mortality. Thus, these particular laboratory parameters reflect a broad range of pathophysiological information within individuals, and the combined use of these biomarkers therefore has the potential for estimating the risk of mortality in a comprehensive and objective manner.

To date, several studies have reported on the combined use of laboratory-only parameters in risk models for in-hospital mortality in patients with ACS, including STEMI [29,30,31,32,33]. Compared with models that use novel biomarkers, such as copeptin and suppression of tumorigenicity 2 [31,32], our model should be easy to perform in the actual clinical setting. Yanishi et al. [33] also reported a simple risk stratification model that was based on the combined use of laboratory parameters to predict in-hospital mortality in Japanese patients with STEMI. In that study, the authors found that a risk-weighted combination model containing the 5 following variables: WBC count and levels of hemoglobin, CRP, creatinine, and BS, could predict that risk. The performance of their model was comparable to that of the TIMI risk score model [34]. Although the actual reasons for the some of the different variables identified by Yanishi et al. versus those identified for our model are unclear, differences between the study cohorts might in part play a role. For example, the study by Yanishi et al. included all patients with STEMI, regardless of whether they were receiving primary PCI and regardless of onset-to-door time. We excluded those patients who did not undergo primary PCI or were admitted >48 h after onset. The rates of in-hospital death were also relatively different between the 2 models (Yanishi’s model 10.8% and our model 5.6%), suggesting that the difference between the extents of severity in study populations affected the selection of variables.

BNP was associated with in-hospital death in the univariate analysis, but not in the multivariate logistic regression. Because potential risk markers were eliminated by multivariate logistic regression using stepwise factor elimination method in the present study, it might be difficult to determine a precise reason for elimination of BNP, as well as other parameters eliminated in the multivariate regression analysis. Measurement of BNP is widely known to improve risk stratification for mortality in patients with STEMI beyond baseline clinical variables [17,35]. On the basis of previous evidence [36], the time-course of the plasma BNP levels in patients with AMI could be divided into two patterns: a monophasic pattern with one peak at about 16 h after admission and a biphasic pattern with two peaks at about 16 h after admission, reflecting acute response to ischemia, and 5 days after admission, reflecting increased wall stress due to cardiac remodeling. The biphasic pattern was associated with severe left ventricular damages and dysfunction. In addition, Suzuki et al. [37] reported that the plasma BNP levels obtained 3 to 4 weeks after the onset of AMI was an independent predictor of cardiac death in patients with AMI. In our study, the BNP levels were measured immediately after admission, and the onset-to-admission time was median 200 min. Thus, the BNP levels obtained in our study unlikely reached at the clinically meaningful levels, and it might be too short and low to predict in-hospital mortality. 

Our laboratory-based risk model was found to provide an accuracy for predicting in-hospital mortality that was similar to that provided by the conventional GRACE risk model. Additionally, our model could further stratify that risk, especially for high-risk patients as identified by the GRACE score. The GRACE score is known to provide several types of information for risk stratification upon admission, including prediction of risk of not only in-hospital mortality, but also 6-month mortality after discharge [38,39]. Moreover, in a Japanese registry of STEMI patients, during a median follow up of 3.9 years for patients with GRACE scores <100, 101–120, 121–140, and ≥141, the mortality rates were 2.0%, 6.3%, 11.8%, and 16.8%, respectively [40]. These results suggest that the GRACE score is useful for predicting the outcomes of patients after STEMI, even during the chronic phase. However, whether or not our laboratory-based model can also predict long-term outcomes is currently unknown. Therefore, further research is warranted to assess the predictive power of our model for long-term outcome.

This study has limitations. First, this was a single-center, retrospective, observational study of a relatively small sample. Second, our inclusion of only those patients with STEMI who underwent primary PCI and were admitted within 48 h after onset could result in bias. Furthermore, the rate of in-hospital mortality was relatively low. Thus, our findings might not be applicable to STEMI patients with other clinical scenarios and different extents of severity. Importantly, we only enrolled Japanese patients, and therefore our findings might only be applicable to this population. Third, this study evaluated laboratory parameters only upon admission for STEMI. Thus, determining if the values of the parameters reflect acute pathophysiological changes associated with STEMI or associated with background status is difficult. In addition, the laboratory parameters obtained after primary PCI, such as lactate and maximum level of CK, were not considered for the current model due to the concept of the study. Finally, we did not take into account any other non-laboratory clinical factors for the risk stratification of in-hospital mortality, including age and sex, cardiac function, duration of ischemia, and perioperative complications. Especially, it has been reported that there were some sex-differences in outcomes after AMI and their predictive markers [41,42,43], however, we could not tease out any sex-dependent issues due to a limited proportion of female population in our cohorts. Although our risk score model showed a performance comparable to that of the GRACE score, further research is thus needed to assess whether the other clinical factors could increase the predictive value for outcome in patients with STEMI.

## 5. Conclusions

The novel laboratory-based model developed for our study cohorts may be helpful for providing an objective and accurate prediction of the risk of in-hospital mortality in STEMI patients who undergo primary PCI within 48 h after onset. The overall accuracy of our model for predicting in-hospital mortality was comparable to that of the conventional GRACE risk model, and our model could stratify that risk further, especially for high-risk patients.

## Figures and Tables

**Figure 1 jcm-09-00852-f001:**
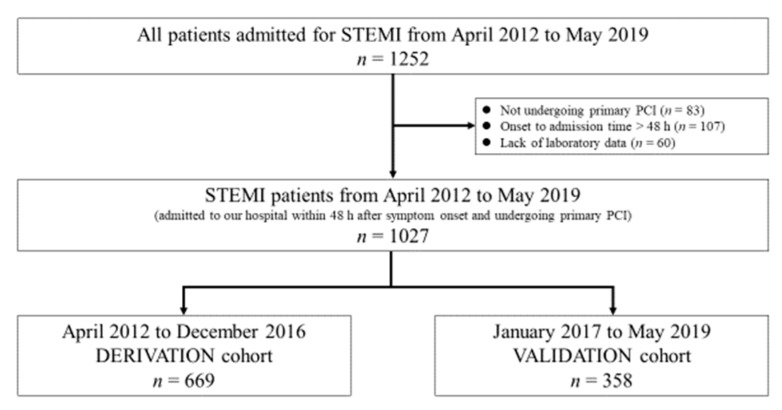
Flow diagram of patient enrollment in the study. PCI, percutaneous coronary intervention; STEMI, ST-segment elevation myocardial infarction.

**Figure 2 jcm-09-00852-f002:**
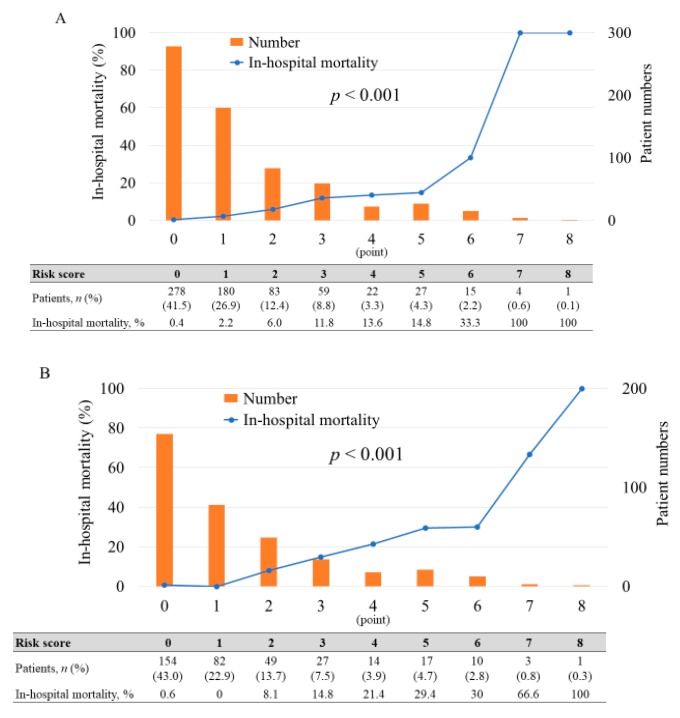
Distribution of risk scores and risk of in-hospital mortality (**A**) derivation cohort and (**B**) validation cohort.

**Figure 3 jcm-09-00852-f003:**
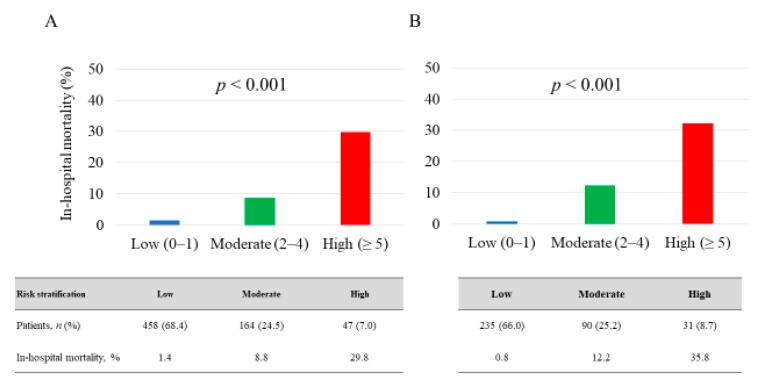
Risk stratification for predicting in-hospital mortality. (**A**) derivation cohort and (**B**) validation cohort. The low-, moderate-, and high-risk groups have scores assigned as follows: 0–1, 2–4, and ≥5, respectively.

**Figure 4 jcm-09-00852-f004:**
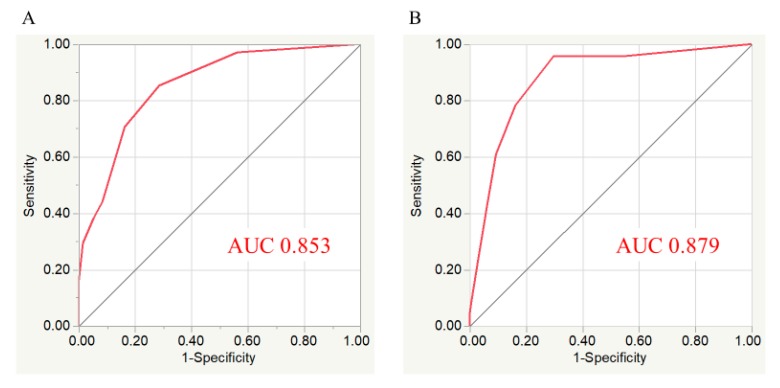
Receiver operating characteristic (ROC) curves of laboratory risk score. (**A**) Area under the curve (AUC) was 0.853 (95% confidence interval (CI) 0.782–0.904) for derivation cohort. (**B**) AUC was 0.879 (95% CI 0.791–0.933) for validation cohort.

**Figure 5 jcm-09-00852-f005:**
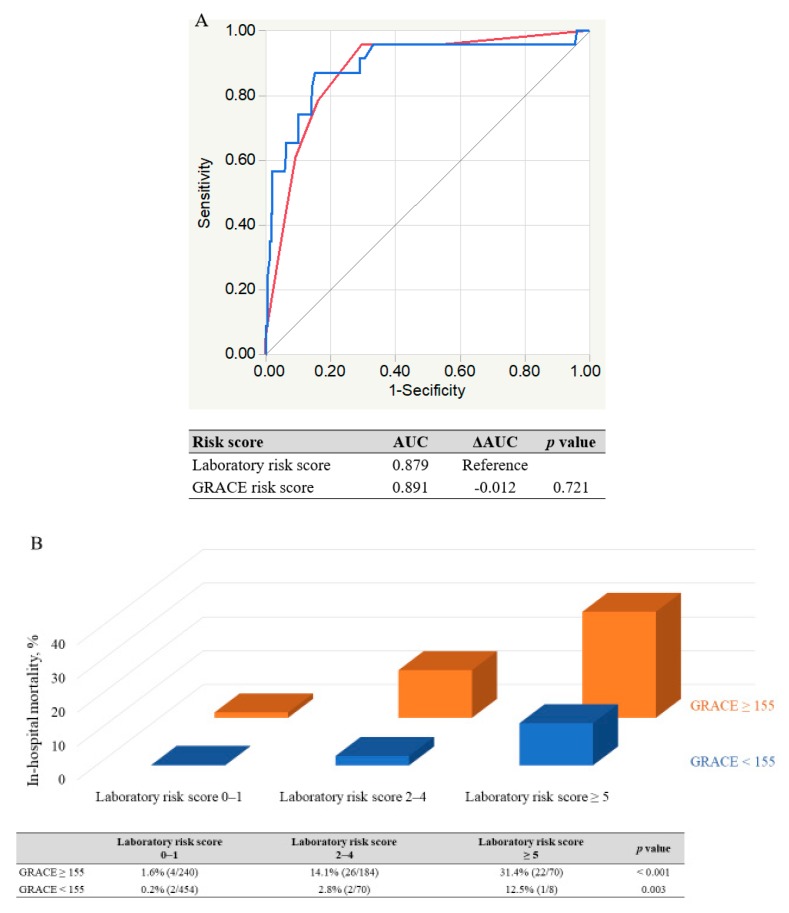
Comparison between laboratory risk score and Global Registry of Acute Coronary Events (GRACE) risk score. (**A**) Area under the curves (AUCs) of laboratory risk score (red) and GRACE risk score (blue) in validation cohort were 0.879 (95% confidence interval (CI) 0.790–0.931) and 0.891 (95% CI 0.773–0.952), respectively, and the difference was not significant. (**B**) Substratification by the combined laboratory parameter risk scores of all patients stratified according to GRACE risk scores. Table below graph shows in-hospital mortality for each stratified population. AUC, area under the curve; CI, confidence interval; GRACE, Global Registry of Acute Coronary Events.

**Table 1 jcm-09-00852-t001:** Baseline characteristics of patients subdivided into derivation and validation cohorts.

Variables	Derivation Cohort (*n* = 669)	Validation Cohort (*n* = 358)	*p*-Value
Age, yr	68.5 ± 12.6	68.9 ± 13.0	0.738
Male, *n* (%)	493 (73.7)	257 (71.8)	0.297
Body mass index, kg/m^2^	23.8 ± 3.6	24.0 ± 3.9	0.228
Systolic BP, mm Hg	138.2 ± 33.4	134.9 ± 32.9	0.488
Heart rate, beats/min	77.0 ± 21.4	76.7 ± 21.1	0.583
Medical history			
Hypertension, *n* (%)	459 (68.6)	232 (64.8)	0.161
Dyslipidemia, *n* (%)	352 (52.6)	185 (51.6)	0.472
Diabetes mellitus, *n* (%)	192 (28.7)	114 (31.8)	0.121
Smoking, *n* (%)	328 (49.0)	198 (55.3)	0.032
Previous MI, *n* (%)	59 (8.8)	28 (7.8)	0.297
Previous PCI, *n* (%)	69 (10.3)	37 (10.3)	0.777
Previous CABG, *n* (%)	7 (1.0)	3 (0.8)	0.777
Laboratory data			
WBC, ×10^2^/μL	106.7 ± 38.0	104.4 ± 37.3	0.472
Hemoglobin, g/dl	13.9 ± 2.2	13.9 ± 2.1	0.885
Platelet, ×10^4^/μL	21.9 ± 6.6	22.0 ± 5.8	0.459
HbA1c, %	5.9 (5.6–6.5)	6.0 (5.6–6.5)	0.235
BS, mg/dL	156 (130–200)	156 (130–200)	0.275
eGFR, mL/min/1.73m^2^	67.4 ± 24.1	65.2 ± 21.3	0.334
LDL-CHO, mg/dL	121.1 ± 35.7	122.6 ± 34.8	0.685
HDL-CHO, mg/dL	46.1 ± 11.6	47.4 ± 12.8	0.577
Albumin, mg/dL	4.0 ± 0.5	4.0 ± 0.5	0.877
Uric acid, mg/dL	6.0 ± 1.6	5.8 ± 1.5	0.236
CRP, mg/dL	0.13 (0.06–0.42)	0.13 (0.06–0.39)	0.303
CK, U/L	154 (96–411)	162 (97–413)	0.581
High-sensitivity troponin I, ng/mL (upper limit of normal: 0.032)	0.27 (0.04–2.64)	0.26 (0.05–2.58)	0.529
BNP, pg/mL (upper limit of normal: 18.4)	47.0 (18.2–151.7)	45.1 (17.2–152.5)	0.401
Killip classification			
I, II, III, IV, *n*	535/74/25/35	292/28/15/22	0.319
Onset-to-admission time, min	200 (110–400)	200 (115–392)	0.381
LVEF (on admission), %	50.6 ± 11.5	48.9 ± 10.0	0.321
Culprit lesion			
LMT, *n* (%)	20 (2.9)	5 (1.4)	0.084
LAD, *n* (%)	348 (52.0))	199 (55.5)	0.110
RCA, *n* (%)	243 (36.3)	117 (32.7)	0.114
LCX, *n* (%)	57 (8.5)	37 (10.3)	0.252
Multi-vessel disease, *n* (%)	269 (38.4)	125 (34.9)	0.278
Pre TIMI grade 0.1, *n* (%)	440 (65.7)	239 (66.9)	0.401
Post TIMI grade 3, *n* (%)	625 (93.4)	323 (90.5)	0.090
Peak CK level, mg/dL	2062 (932–3899)	2096 (973–3934)	0.350
Mechanical support on admission			
Respirator, *n* (%)	36 (5.4)	18 (5.0)	0.436
Temporary pacing, *n* (%)	51 (7.6)	33 (9.2)	0.238
IABP, *n* (%)	99 (14.7)	44 (12.3)	0.169
PCPS, *n* (%)	21 (3.1)	11 (3.0)	0.523
In-hospital death, *n* (%)	34 (5.1)	23 (6.4)	0.318

Data for categorical variables are given as numbers (%); data for continuous variables given as means ± standard deviation for normal distribution or medians (interquartile range) for skewed distribution. BNP, brain natriuretic peptide; BP, blood presser; BS, blood sugar; CABG, coronary artery bypass grafting; CK, creatine kinase; CRP, C-reactive protein; eGFR, estimated glomerular filtration rate; HDL-CHO, high density lipoprotein cholesterol; HR, heart rate; IABP, intra-aortic balloon pumping; LAD, left anterior descending; LCX, left circumflex; LDL-CHO, low density lipoprotein cholesterol; LMT, left main tank; LVEF, left ventricular ejection fraction; MI, myocardial infarction; PCI, percutaneous coronary intervention; PCPS, percutaneous cardiopulmonary support; RCA, right coronary artery; TIMI, thrombolysis in myocardial infarction; WBC, white blood cell.

**Table 2 jcm-09-00852-t002:** Univariate analysis for in-hospital death in derivation cohort.

Variables	Survivor	Death	*p*-Value
WBC, ×10^2^ /μL	105.9 ± 1.5	122.6 ± 6.5	0.013
Hemoglobin, g/dL	14.0 ± 2.1	12.6 ± 2.1	<0.001
Platelet, ×10^4^/μL	22.1 ± 0.6	17.3 ± 1.1	<0.001
HbA1c, %	5.9 (5.7–6.5)	6.1 (5.6–6.9)	0.541
BS, mg/dL	155 (129–194)	200 (159–291)	<0.001
eGFR, mL/min/1.73m^2^	68.5 ± 0.9	46.1 ± 4.0	<0.001
LDL-CHO, mg/dL	125.5 ± 1.4	117.3 ± 6.1	0.191
HDL-CHO, mg/dL	44.6 ± 0.5	43.5 ± 2.0	0.182
Albumin, mg/dL	4.1 ± 0.1	3.4 ± 0.1	<0.001
Uric acid, mg/dL	5.9 ± 0.1	7.4 ± 0.3	<0.001
CRP, mg/dL	0.13 (0.06–0.39)	0.35 (0.08–1.06)	0.012
CK, U/L	152 (96–393)	312 (133–2148)	0.005
High-sensitivity troponin I, ng/mL (99th percentile for whole healthy adult population: 0.026)	0.26 (0.04–2.18)	2.32 (0.29–40.4)	<0.001
BNP, pg/mL	44.9 (16.8–138.0)	240 (55.4–805)	<0.001

Data for continuous variables are given as means ± standard deviation for normal distribution or medians (interquartile range) for skewed distribution. BNP, brain natriuretic peptide; BS, blood sugar; CK, creatine kinase; CRP, C-reactive protein; eGFR, estimated glomerular filtration rate; HDL-CHO, high density lipoprotein cholesterol; LDL-CHO, low density lipoprotein cholesterol; WBC, white blood cell.

**Table 3 jcm-09-00852-t003:** Multivariate logistic regression in derivation cohort and corresponding risk score for in-hospital death.

Variables	Odds Ratio	95% Confidence Interval	*p*-Value	Given Score
Platelet <15 × 10^4^/μL	3.45	1.50–7.97	0.003	2 points
BS ≥200 mg/dL	2.63	1.20–5.80	0.020	1 point
eGFR <45 mL/min/1.73m^2^	3.65	1.60–8.11	0.001	2 points
Albumin ≤3.5mg/dL	3.37	1.52–7.47	0.003	2 points
High-sensitivity troponin I >1.6 ng/dL (normal upper limit × 50)	2.76	1.27–6.01	0.010	1 point

BS, blood sugar; eGFR, estimated glomerular filtration rate.

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
