# Peer review of "A Novel Predictive Model for In-Hospital Mortality Based on a Combination of Multiple Blood Variables in Patients with ST-Segment-Elevation Myocardial Infarction"

_jcm, 2020, doi:10.3390/jcm9030852_

Round 1

Reviewer 1 Report

The authors present a predictive model for in-hospital mortality based on a combination of different blood values in STEMI patients. Data was collected retrospectively in one center. Indeed it is appealing to calculate mortality risk solely on the basis of objective blood values. Needless to say that there are numerous scores available to calculate the risk of STEMI patients.  

Major issue:

  1. Of course you know that cardiogenic shock is the most frequent cause of death for acute myocardial infarction patients in hospital. Was lactate measured and included in the analysis?
  2. Was maximal level of CK included into univariate analysis?

Author Response

Response to Reviewer 1 Comments

Firstly, we are very grateful for the opportunity of revising the manuscript. We have carefully revised the manuscript according to your valuable scientific comments. We hope that our revised manuscript would meet the publication standard of Journal of Clinical Medicine.

Point 1: Of course you know that cardiogenic shock is the most frequent cause of death for acute myocardial infarction patients in hospital. Was lactate measured and included in the analysis?

Response 1: Thank you for your important comment. We completely agree with your comment on the cardiogenic shock as the most frequent cause of in-hospital death after acute myocardial infarction and its relationship with increased level of lactate. However, unfortunately, because the lactate was not measured routinely on admission (= immediately after admission) in this study, that was not included in the analysis. In addition, because the concept of this study was to assess whether we could develop a prediction model for in-hospital mortality based on a combination of blood test parameters obtained on admission (= immediately after admission), the laboratory parameters obtained after primary PCI were not considered for the current model. Therefore, we have added this issue in the limitation (see Page 11, Line 311-313).    

Point 2: Was maximal level of CK included into univariate analysis?

Response 2: Thank you for your important comment. We also acknowledge that the maximum level of CK should be an important risk factor of in-hospital death after STEMI. However, the maximum level of CK would be generally proved after primary PCI, but not on admission (= immediately after admission). As described above response 1, on the basis of concept of this study we did not include that level into our model. Nonetheless, as an important limitation of our study, we have added this issue in the limitation (see Page 11, Line 311-313).

Reviewer 2 Report

This is a useful study examining the performance of a panel of biomarkers in predicting in-hospital mortality in a small cohort of patients presenting with STEMI. Although observational and retrospective in nature, your study contributes insight into the utility of eGFR, hsTnI, platelet count, blood sugar level and albumin in predicting death post-STEMI in the short term. Exclusions were appropriate (those not receiving primary PCI, those with late onset to admission (time > 48 hours) and those with absent blood tests), and though your study is relatively small, it was sufficiently powered to test the performance of your algorithm. The use of a biomarker panel that can be quickly derived early in presentation of STEMI is appealing, particularly as the markers you have assessed are mainstream laboratory tests, not novel or inaccessible biomarkers. The paper can be further improved upon through attention to the following (small) details:

Methodology: Please include the platforms/methods used for the laboratory testing. This is particularly relevant for hsTnI: from the dates of your study, I presume the Abbott high sensitivity troponin I assay was used? Though you have expressed data as continuous variables, it would be useful to also include reference intervals for each biomarker. Again for troponin, the reader needs to see the 99th centile used for interpretation in your hospital, so we can understand the survivor and death median hsTnI results relative to this 99th percentile (presumably overall, not sex specific?). More broadly, we need to know other laboratory test details. For example, what method was used for HbA1c determination? For BNP?

Sex-dependent differences: The ways in which sex influences outcomes after AMI is topical and I think your paper would benefit from consideration of this, particularly as some of the laboratory parameters have sex-specific reference intervals (including hsTnI). Your cohort is, as expected, more than 70% male, and perhaps you do not have enough females included in order to tease out sex-dependent issues. Nonetheless this issue deserves at least comment within your manuscript.

Timing of samples: You rightly point out that BP and heart rate, components of the GRACE score, can vary widely during acute presentation of STEMI. Yet biomarkers do, too. Can you please give an indication of the time at which bloods were taken during presentation? It might be important to show that the higher hsTnI and BNP levels in those patients who died was not due to later presentation/later bloods. I understand that time of onset of symptoms is difficult to pin down, but because you have excluded late presenters (> 48 hours after onset), I presume you have indeed captured this time of onset data. If so, this could easily be combined with timing of laboratory data to estimate time between onset and

A couple of smaller questions:

BNP was associated with death in univariate analysis, but not in the multivariate logistic regression. What is your interpretation of this?

Blood glucose concentration was associated with mortality in both univariate and multivariate analyses, but not HbA1c. Do you think this reflects acute hyperglycaemia associating with mortality (eg. as a reflection of a stress response?), rather than underlying, longer term diabetes status, which may be a risk factor for AMI, but according to your study, is not a predictor of short term death.

Author Response

Response to Reviewer 2 Comments

Firstly, we are very grateful for the opportunity of revising the manuscript. We have carefully revised the manuscript according to your valuable scientific suggestions. We hope that our revised manuscript would meet the publication standard of Journal of Clinical Medicine.

Point 1: Methodology: Please include the platforms/methods used for the laboratory testing. This is particularly relevant for hsTnI: from the dates of your study, I presume the Abbott high sensitivity troponin I assay was used? Though you have expressed data as continuous variables, it would be useful to also include reference intervals for each biomarker. Again for troponin, the reader needs to see the 99th centile used for interpretation in your hospital, so we can understand the survivor and death median hsTnI results relative to this 99th percentile (presumably overall, not sex specific?). More broadly, we need to know other laboratory test details. For example, what method was used for HbA1c determination? For BNP?

Response 1: Thank you for your helpful suggestion on this methodological issue. In accordance with your suggestion, we have provided the laboratory test details as follows;

hsTnI assay:

ARCHITECT®︎ high sensitive troponin I (Abbott Japan, Tokyo, Japan) on an ARCHITECT® i1000SR analyzer (Abbott Japan, Tokyo, Japan) (see the Methods section ‘2.3. Data Collection’).

Reference interval for hsTnI:

Upper limit of normal is 0.032 ng/mL (see Table 1).

99th percentile of hsTnI for whole healthy adult population:

0.026 ng/mL (see Table 2).

BNP assay:

AIA®︎- 900 analyzer (TOSHO BIOSCIENCE, Tokyo, Japan) (see the Methods section ‘2.3. Data Collection’).

Reference interval for BNP:

Upper limit of normal is 18.4 pg/ml (see Table 1).

The method for WBC:

Sysmex XN-1000TM (SYSMEX CORPORATION, Hyogo, Japan) (see the Methods section ‘2.3. Data Collection’).

The method for platelet:

Sysmex XN-1000TM (SYSMEX CORPORATION, Hyogo, Japan) (see the Methods section ‘2.3. Data Collection’).

The method for hemoglobin:

Sysmex XN-1000TM (SYSMEX CORPORATION, Hyogo, Japan) (see the Methods section ‘2.3. Data Collection’).

The method for CRP:

BioMajestyTM Series JCA-BM6010 (JEOL Ltd., Tokyo, Japan) (see the Methods section ‘2.3. Data Collection’).

The method for creatinine:

BioMajestyTM Series JCA-BM6010 (JEOL Ltd., Tokyo, Japan) (see the Methods section ‘2.3. Data Collection’).

The method for CK:

BioMajestyTM Series JCA-BM6010 (JEOL Ltd., Tokyo, Japan) (see the Methods section ‘2.3. Data Collection’).

The method for BS:

BioMajestyTM Series JCA-BM6010 (JEOL Ltd., Tokyo, Japan) (see the Methods section ‘2.3. Data Collection’).

The method for HbA1c:

HLC-723®︎G9 analyzer (TOSHO BIOSCIENCE, Tokyo, Japan) (see the Methods section ‘2.3. Data Collection’).

The method for albumin:

BioMajestyTM Series JCA-BM6010 (JEOL Ltd., Tokyo, Japan) (see the Methods section ‘2.3. Data Collection’).

The method for UA:

BioMajestyTM Series JCA-BM6010 (JEOL Ltd., Tokyo, Japan) (see the Methods section ‘2.3. Data Collection’).

The method for LDL-CHO:

BioMajestyTM Series JCA-BM6010 (JEOL Ltd., Tokyo, Japan) (see the Methods section ‘2.3. Data Collection’).

The method for HDL-CHO:

BioMajestyTM Series JCA-BM6010 (JEOL Ltd., Tokyo, Japan) (see the Methods section ‘2.3. Data Collection’).

Point 2: Sex-dependent differences: The ways in which sex influences outcomes after AMI is topical and I think your paper would benefit from consideration of this, particularly as some of the laboratory parameters have sex-specific reference intervals (including hsTnI). Your cohort is, as expected, more than 70% male, and perhaps you do not have enough females included in order to tease out sex-dependent issues. Nonetheless this issue deserves at least comment within your manuscript.

Response 2: Thank you very much for your kind suggestion on this important issue. As you stated, the sex-dependent differences in the outcomes after ischemic heart diseases, including AMI, should be topical and very important clinical issue. In addition, some of the laboratory parameters used in our study also have sex-specific reference intervals, including hsTnI. However, because the concept of this study was to assess whether we could develop a prediction model for in-hospital mortality only based on a combination of blood test parameters obtained on admission, we did not take into account any other non-laboratory clinical factors, including age and sex, in our risk–score model. In addition, as you also noticed, because a large proportion of our cohort was male (more than 70%), it would be not enough to tease out sex-dependent issues. Therefore, we have discussed and described this issue in the limitations section (see Page 11, Line 313-318).  

Point 3: Timing of samples: You rightly point out that BP and heart rate, components of the GRACE score, can vary widely during acute presentation of STEMI. Yet biomarkers do, too. Can you please give an indication of the time at which bloods were taken during presentation? It might be important to show that the higher hsTnI and BNP levels in those patients who died was not due to later presentation/later bloods. I understand that time of onset of symptoms is difficult to pin down, but because you have excluded late presenters (> 48 hours after onset), I presume you have indeed captured this time of onset data. If so, this could easily be combined with timing of laboratory data to estimate time between onset and (authors could not find any continuation of this sentence)

Response 3: Thank you for your important and helpful comment, In the present study, an indication of the time when bloods were taken was immediately after admission, and this was also a clinical routine in our emergency department. Therefore, we have changed ‘‘admission blood tests’’ into ‘‘blood tests parameters obtained immediately after admission’’ in the Abstract (see Page 1, Line 22-23); ‘‘on admission’’ into ‘‘immediately after admission’’ in the Methods section ‘2.3. Data Collection’ (see Page 3, Line 96).

   In addition, there was no significant difference in the onset-to-admission time between in-hospital survival group and death group in overall cohort (survivor, 200 min [interquartile range 115–385] vs. death, 260 min [145–630], p = 0.062), suggesting the higher hsTnI and BNP levels in the death group were unlikely due to later presentation and/or sampling. According to your suggestion, we have described this result in the Results section ‘3.1. Patient Characteristics’ (see Page 4, Line 139-142) and discussed this important issue in the Discussion section (see Page 10, Line 235-238).

Point 4: BNP was associated with death in univariate analysis, but not in the multivariate logistic regression. What is your interpretation of this?

Response 4: Thank you for pointing out this clinically important point. First of all, because potential risk markers were eliminated by multivariate logistic regression using stepwise factor elimination method in the present study, it might be difficult to determine a precise reason for elimination of BNP, as well as other parameters eliminated in the multivariate regression analysis. Measurement of BNP is widely known to improve risk stratification for mortality in patients with STEMI beyond baseline clinical variables [Mega et al. J Am Coll Cardiol 2004; 44: 335-339, Scirica et al. Clin Chem 2013; 59: 1205-1214]. On the basis of previous evidence [Morita et al. Circulation 1993; 88: 82-91], the time-course of the plasma BNP levels in patients with AMI could be divided into two patterns: a monophasic pattern with one peak at about 16 hours after admission and a biphasic pattern with two peaks at about 16 hours after admission, reflecting acute response to ischemia, and 5 days after admission, reflecting increased wall stress due to cardiac remodeling. The biphasic pattern was associated with severe left ventricular damages and dysfunction. In addition, Suzuki et al. [Circulation 2004; 110: 1387-1391] reported that the plasma BNP levels obtained 3 to 4 weeks after the onset of AMI was an independent predictor of cardiac death in patients with AMI. In our study, the BNP levels were measured immediately after admission, and the onset-to-admission time was median 200 min. Thus, the BNP levels obtained in our study unlikely reached at the clinically meaningful levels, and it might be too short and low to predict in-hospital mortality. We have described these sentences in the Discussion section (see Page 11, Line 275-290). Thank you very much for your valuable comment again.           

Point 5: Blood glucose concentration was associated with mortality in both univariate and multivariate analyses, but not HbA1c. Do you think this reflects acute hyperglycaemia associating with mortality (eg. as a reflection of a stress response?), rather than underlying, longer term diabetes status, which may be a risk factor for AMI, but according to your study, is not a predictor of short term death.

Response 5: Thank you for your important comment. As you stated, we also completely agree with your opinion. We have discussed this interesting issue in the Discussion section (see Page 10, Line 251-255) in accordance with your comment.   

Round 2

Reviewer 1 Report

Answers are sufficient for me.